# *APOB* CRISPR-Cas9 Engineering in Hypobetalipoproteinemia: A Promising Tool for Functional Studies of Novel Variants

**DOI:** 10.3390/ijms23084281

**Published:** 2022-04-13

**Authors:** Xavier Vanhoye, Alexandre Janin, Amandine Caillaud, Antoine Rimbert, Fabienne Venet, Morgane Gossez, Wieneke Dijk, Oriane Marmontel, Séverine Nony, Charlotte Chatelain, Christine Durand, Pierre Lindenbaum, Jennifer Rieusset, Bertrand Cariou, Philippe Moulin, Mathilde Di Filippo

**Affiliations:** 1Service de Biochimie et de Biologie Moléculaire, Laboratoire de Biologie Médicale MultiSites, Hospices Civils de Lyon, F-69677 Bron, France; xavier.vanhoye@chu-lyon.fr (X.V.); alexandre.janin01@chu-lyon.fr (A.J.); oriane.marmontel@chu-lyon.fr (O.M.); severine.nony@chu-lyon.fr (S.N.); charlotte.chatelain@chu-lyon.fr (C.C.); 2Institut NeuroMyoGène, CNRS UMR5310, INSERM U1217, Université Claude Bernard Lyon 1, Université de Lyon, F-69008 Lyon, France; 3Institut du Thorax, Nantes Université, CHU Nantes, CNRS, INSERM, F-44000 Nantes, France; amandine.caillaud@univ-nantes.fr (A.C.); bertrand.cariou@univ-nantes.fr (B.C.); 4Institut du Thorax, Nantes Université, CNRS, INSERM, F-44000 Nantes, France; antoine.rimbert@univ-nantes.fr (A.R.); wieneke.dijk@inrae.fr (W.D.); pierre.lindenbaum@univ-nantes.fr (P.L.); 5Laboratoire d’Immunologie, Edouard Herriot Hospital, Hospices Civils de Lyon, F-69437 Lyon, France; fabienne.venet@chu-lyon.fr (F.V.); morgane.gossez@chu-lyon.fr (M.G.); 6Centre International de Recherche en Infectiologie (CIRI), INSERM U1111, CNRS, UMR5308, Ecole Normale Supérieure de Lyon, Université Claude Bernard-Lyon 1, F-69364 Lyon, France; 7CarMen Laboratory, INSERM, INRA, INSA Lyon, Université Claude Bernard Lyon 1, Pierre-Bénite, F-69364 Lyon, France; christine.durand@univ-lyon1.fr (C.D.); jennifer.rieusset@univ-lyon1.fr (J.R.); philippe.moulin@chu-lyon.fr (P.M.); 8Fédération d’Endocrinologie, Maladies Métaboliques, Diabète et Nutrition, Hôpital Louis Pradel, Hospices Civils de Lyon, F-69677 Bron, France

**Keywords:** primary hypobetalipoproteinemia, apolipoprotein B, genome editing, variants of uncertain significance, functionality, secretion, cholesterol

## Abstract

Hypobetalipoproteinemia is characterized by LDL-cholesterol and apolipoprotein B (apoB) plasma levels below the fifth percentile for age and sex. Familial hypobetalipoproteinemia (FHBL) is mostly caused by premature termination codons in the *APOB* gene, a condition associated with fatty liver and steatohepatitis. Nevertheless, many families with a FHBL phenotype carry *APOB* missense variants of uncertain significance (VUS). We here aimed to develop a proof-of-principle experiment to assess the pathogenicity of VUS using the genome editing of human liver cells. We identified a novel heterozygous *APOB*-VUS (p.Leu351Arg), in a FHBL family. We generated *APOB* knock-out (KO) and *APOB*-p.Leu351Arg knock-in Huh7 cells using CRISPR-Cas9 technology and studied the *APOB* expression, synthesis and secretion by digital droplet PCR and ELISA quantification. The *APOB* expression was decreased by 70% in the heterozygous *APOB*-KO cells and almost abolished in the homozygous-KO cells, with a consistent decrease in apoB production and secretion. The *APOB*-p.Leu351Arg homozygous cells presented with a 40% decreased *APOB* expression and undetectable apoB levels in cellular extracts and supernatant. Thus, the p.Leu351Arg affected the apoB secretion, which led us to classify this new variant as likely pathogenic and to set up a hepatic follow-up in this family. Therefore, the functional assessment of *APOB*-missense variants, using gene-editing technologies, will lead to improvements in the molecular diagnosis of FHBL and the personalized follow-up of these patients.

## 1. Introduction

Hypobetalipoproteinemia (HBL) is characterized by low plasma levels of apolipoprotein B (apoB) and low-density lipoprotein cholesterol (LDL-C), less than the fifth percentile for age and sex [1]. In total, 30 to 50% of patients with primary HBL present a polygenic origin of HBL [2,3], whereas a monogenic origin is identified in 20 to 50% of patients [4]. The most common etiology of monogenic HBL is familial HBL1 (FHBL1 (MIM: 615,558)), a semi-dominant variant in the *APOB* gene (MIM: 107,730), leading to apoB secretion defects.

ApoB plays a central role in the transport and metabolism of cholesterol and triglycerides (TG) in plasma. The tissue-specific differential mRNA processing of the *APOB* gene leads to two different apoB isoforms. ApoB-48, essential for the absorption of exogenous lipids through chylomicron production, produced in the intestine, results from a premature termination codon (PTC) at the codon 2153 [5]. The liver-produced full-length apoB (apoB-100) is instead an essential component of very-low-density lipoproteins (VLDL) and LDL [6] and binds to the LDL receptor (LDLR), allowing LDL clearance [7]. With its dual role, apoB is therefore involved in hypercholesterolemia in cases of LDLR-binding defects (MIM: 144,010) [7,8] and in FHBL1 in cases of apoB secretion defects. Over 200 deleterious variants producing mostly apoB truncations, ranging from 2 to 89% of full-length protein, have been identified as causes of FHBL1 [9]. Rare missense variants located in the βα1 domain of both apoB-48 and apoB-100 [10] have been found in HBL patients [11,12,13,14]. This domain is involved in microsomal transfer protein (MTP) binding [15] and in apoB lipidation; this is due to specific physio-chemical properties allowing the formation of a “lipid pocket” [16,17,18]. In vitro studies have shown that several of these variants impair the secretion of apoB-48 or smaller apoB isoforms in transfected cells and, thus, cause FHBL [11,12,13].

Carrying a heterozygous *APOB* (He-*APOB*) PTC was associated with a significantly lower risk for coronary heart disease [19]. However, unlike other causes of primary HBL, this condition is associated with an increased risk of NAFLD (non-alcoholic fatty liver disease) when compared to the general population [3,20,21]. NASH (non-alcoholic steatohepatitis) [19], cirrhosis and hepatocellular carcinoma have been described in He-*APOB* [14,22,23,24]. Patients carrying early premature truncations of apoB seem to have a more pronounced NAFLD phenotype [25] than patients with late truncations or with single-amino-acid changes in the apoB protein [4]. However, some patients carrying He-*APOB* missense variants have developed NASH and liver cirrhosis [14,26]. This highlights the clinical value of determining whether the new *APOB* rare missense variants found in HBL patients are causative.

We here identified p.Leu351Arg, a rare *APOB* missense variant in a family presenting with a FHBL1 phenotype. Despite its co-segregation with HBL in the proband and its two children, this variant was classified as a variant of uncertain significance (VUS) according to the guidelines of the American College of Medical Genetics and Association of Medical Pathologists (ACMG) [27]. To test whether the FHBL phenotype resulted from p.Leu351Arg, we edited hepatic cells (Huh7) by CRISPR-Cas9 engineering and tested the impact of this variant on the full length apoB-100 hepatic synthesis and secretion. A positive result on this test (absence of ApoB-100 secretion) would allow the reclassification of this variant as a likely pathogenic (in class 4, according to the ACMG guidelines).

## 2. Results

### 2.1. Clinical Phenotyping and Genotyping 

After a routine lipid profiling, performed in the context of her diabetes follow-up, we identified a patient with a clinical diagnosis of HBL (I.1—Table 1, Figure 1). The HBL complementary checkup revealed an S3 steatosis score and mild vitamin E deficiency. After familial investigations, her son (II.1) and her daughter (II.2) were also found to present with HBL. Her son also exhibited liver cytolysis and a mild liposoluble-vitamin deficiency was found in her daughter (II.2), but not in her son (II.1). The family phenotype is described in Table 1 and Figure 1.

### 2.2. Identification of APOB Variant

A rare heterozygous missense variant in the *APOB* gene, NM_000384.2: c.1052T > G, p.(Leu351Arg) (called Leu351Arg), was identified in the proband (I.1, Figure 1). The performance of Sanger sequencing on the family showed that *APOB-*Leu351Arg co-segregates with the HBL phenotype (Figure 1). 

The Leu351Arg variant is located in exon 9 of the *APOB* gene. This variant was not detected in the general population (GnomAD v2.1.1), nor in Clinvar and HGMD patient cohorts. This variant affects the same amino acid as the previously reported pathogenic variant p.(Leu351Met) (called Leu351Met) [13] and was predicted to be pathogenic by in silico software (CADD: 3.969, phred-scaled = 26.8). No effect on splicing was predicted. This variant co-segregated with the HBL phenotype in the family, but the small number of family members precluded the feasibility of performing a genetic linkage analysis. Following the ACMG guidelines [27], this variant was initially classified as being of uncertain significance.

Using whole-genome sequencing (WGS) analysis, we screened for other variants in known genes (namely *ANGPTL3*, *APOB*, *MTTP*, *PCSK9* and *SAR1B*) that could be responsible for the HBL phenotype. No rare single-nucleotide variant and no copy-number variation were found in the genes involved in FHBL. The polygenic risk score (PRS) did not support a polygenic HBL (I.1 PRS = 1.034 (70th–75th percentile of controls); II.1 PRS = 1.043 (70–75th percentile); and II.2 PRS = 1.057 (75–80th percentile)). We thus hypothesized that the *APOB* Leu351Arg variant might be responsible for the FHBL phenotype observed in this family.

### 2.3. Leu351Arg Modeling

The substitution is located in the apoB βα1 domain involved in apoB-48 and apoB-100 lipidation [16,17,18]. The substitution Leu351Arg replaces a buried hydrophobic, uncharged residue with a hydrophilic, charged residue. The buried wild-type residue is not involved in any H-bonds. Nevertheless, Arg351 can be involved into two H-bonds: one with the main chain of Val325 (3.5 Å) and another with the side chain of Thr378 (3.10 Å) (Figure 2a). This substitution leads to a contraction in the cavity volume of 30.24 Å^3^ (Figure 2b,c).

### 2.4. CRISPR/Cas9 Engineering

To study the effect of this variant on apoB secretion, we introduced the Leu351Arg variant into Huh7 cells, which naturally secrete the full-length apoB-100. A stable cell line carrying Leu351Arg in the homozygous state was obtained. As expected, this clone also carries two synonymous variants: c.1059T > C p.(Thr353=) and c.1066A > C p.(Arg356=), which are not predicted to affect the splicing of *APOB* RNA, nor to be deleterious (CADD = −0.178, PHRED-scaled = 0.827 and CADD = 0.292, PHRED-scaled = 7.118, respectively).

Non-homologous end-joining induces the knock-out (KO) cell lines c.1064dup and p.(Arg356Glufs*5) in homozygous (Ho-KO) and heterozygous states (He-KO). These clones were used as positive controls of FHBL (Appendix A Appendix A).

### 2.5. Leu351Arg Impaired apoB-100 Production and Secretion 

FHBL1 causative variants impair apoB-100 secretion. We therefore set out to determine the impact of the Leu351Arg variant on the secretion of apoB-containing lipoprotein in the medium of Huh7 cells following oleic acid treatment. ApoB-100 secretion was measured in the Huh7 wild-type (WT), Ho-KO, He-KO and Ho-Leu351Arg stable cell lines. When normalized to the protein concentrations in the cell lysate, the concentration of apoB in the medium increased during the incubation time with oleic acid in the Huh7 WT, as previously published [13], increased mildly in the He- KO (60% of WT at T6h, *p*-value = 7 × 10^−2^) and was nearly absent in the medium of both the Ho-KO (*p*-value = 1.2 × 10^−2^) and the Ho-Leu351Arg (*p*-value = 1.2 × 10^−2^) (Figure 3). 

To study whether the Leu351Arg impaired apoB synthesis, the intracellular apoB concentration was measured in the four Huh7 WT, Ho-KO, He-KO and Ho-Leu351Arg stable cell lines. When normalized to the protein concentrations in the cell lysate, the apoB concentrations increased during the incubation time with oleic acid in the Huh7 WT cell line, moderately increased in the He-Ko (60% of WT at T6h, *p*-value = 4.1 × 10^−2^) and were low and stable in the Ho-KO and Ho-Leu351Arg cell lines (*p*-value = 1.2 × 10^−2^) (Figure 4). 

The RT-PCR of the *APOB* transcripts in the Huh7 cells showed that the expression level was nearly absent in Ho-KO, probably resulting from nonsense-mediated decay (Figure 5). The *APOB* mRNA in the He-KO cells decreased compared to the WT (29% of WT, *p*-value = 2.2 × 10^−6^) (Figure 5). Interestingly, the *APOB* mRNA decreased by 40% in the Ho-Leu351Arg cells (*p*-value = 3.5 × 10^−4^) (Figure 5). 

Altogether, these results suggest that the Leu351Arg variant, which is localized within the βα1 domain of apoB, decreases APOB mRNA, drastically inhibits apoB-synthesis and almost entirely blocks apoB secretion. These findings establish the functional impact of Leu351Arg.

## 3. Discussion

Only a few *APOB* missense variants have been described in FHBL1 [11,12,13]. Since *APOB* is a large gene (42 kb), the functionality of these missense variants was previously assessed by the transfection of plasmid-encoding premature truncated proteins, namely apoB-17 or apoB-48. To be as close as possible to human hepatocyte physiology, we chose to use human hepatic cells (Huh7) that secrete the full-length apoB protein. We here show that the apoB-100 secretion in Leu351Arg genome-edited cells was impaired. The present model unambiguously demonstrates that the Leu351Arg variant is FHBL-causative. Moreover, our proof-of-concept study demonstrates the capacity of this model to assess the functionality of any new *APOB* variant, regardless of its location in the gene.

To date, this is the first study providing a cellular model of FHBL1 with an *APOB* KO in heterozygous and homozygous states in hepatic cells. We validated this model by showing the near-absence of *APOB* mRNA and apoB100 in Ho-KO. ApoB proteins less than 30% [28] of the full length were not secreted and the ApoB-100 ELISA assay targeted an epitope not included in the apob-48 domain. 

The *APOB* mRNA and apoB-100 concentrations were lower in the He-KO compared to the WT. Intriguingly, while a 50% decrease in plasma LDL-C and apoB concentrations is expected in FHBL1 patients (with one *APOB*-deficient allele of two) they usually present with a 70% decrease when compared to age- and gender-matched controls [29], suggesting a dominant negative effect [30]. Our *APOB* mRNA results in He-KO reinforce this hypothesis and are consistent with previous studies, showing the cytoplasmic degradation of the PTC carrier *APOB*-mRNA [30,31]. In the Ho-Leu351Arg cells, the 40% *APOB* mRNA decrease probably contributed to the observed decrease in the protein level. The *APOB* sequencing showed that this decrease could not have been caused by the CRISPR/Cas9 off-target or copy-number variants. Moreover, no effect on splicing was predicted by the in silico tools. Finally, other post-transcriptional regulations involving miRNA, long coding RNA or differences in the intracytoplasmic amount of transfer RNA should be explored [32,33,34]. Most FHBL1-causative missense variants are located in the apoB βα1 domain common in apoB48 and apoB100. Here, we report a new variant in this domain: Leu351Arg. We demonstrated that Ho-Leu351Arg cells phenocopy Ho-KO cells for apoB-100 secretion. Since the ELISA kit is specific to apoB100 and does not recognize apoB-48, the lack of signal is not due to the missense variant that occurs in the common domain at apoB100 and apoB48. As FHBL1 is a semi-dominant condition, these results are strong enough to consider this variation as deleterious, even though we failed to generate Leu351Arg heterozygous cells. 

Notably, a variant located in the same amino acid has been previously studied: p.Leu351Met, termed L324M according the nomenclature without the signal peptide [13]. Zhong et al. demonstrated that plasmids containing *APOB17*-Leu351Met and *APOB48*-Leu351Met lead to reduced apoB secretion [13]. ApoB was not detectable in the supernatant of the Ho-Leu351Arg cells, which confirms the essential role of this amino acid in efficient VLDL secretion. Interestingly, whereas Zhong et al. demonstrated no modification of apoB in cells transfected with *APOB17* or *APOB48*-Leu351Met, we found almost no apoB-100 in the cell lysates of the Ho-Leu351Arg cells, which were engineered with CRISPR-Cas. This discrepancy could be due to the size of the apoB protein (17% or 48% of the full-length apoB). Indeed, since it was shown that MTTP bound better to shorter apoB peptides compared to full-length apoB [35], MTTP might have a high affinity for apo17 and apoB48 compared with apoB-100. On the other hand, according to Grantham’s distance, leucine and methionine (Grantham’s distance = 15) are closer than leucine and arginine (Grantham’s distance = 102). This difference is mainly due to hydrophobicity changes: leucine and methionine are hydrophobic, whereas arginine is hydrophilic. Furthermore, this variant occurs in the lipid pocket of apoB, containing hydrophobic interfaces that are supposed to interact with lipids. This α-helical B6.4-13 contains some hydrophobic helices that strongly bind the lipids [17,18]. Thus, we can hypothesize that Leu351Arg has a greater impact on lipid binding than Leu351Met and might increase the impairment of the Mtp-mediated lipidation of apoB-100, leading to enhanced apoB degradation via the ubiquitin-proteasome pathway [36]. The Leu351Arg editing failed in the HuH7 cells; the impact of the two different variants could not be compared using the same model. Moreover, more investigations are needed to evaluate the catabolism of this misfolded apoB and these stable cell lines will be of interest to further characterize the apoB cellular pathway in FHBL models.

This study as some limitations. For example, the impact of the variant was evaluated only by apoB measurement. The additional biological impacts of the altered apoB secretion were not evaluated. An accumulation of TG was expected, but the suppression of lipogenesis was observed for other truncating and non-truncating FHBL-causative variants in vitro [13]. Thus, the evaluation of this biological impact requires the concomitant measurement of the intracellular TG concentration and expression of genes involved in lipogenesis. In addition, the impact of the variant on lipoprotein secretion requires the study of the composition of lipoprotein after isolation and/or pulse-chase analysis. None of these methods are conveniently accessible in a routine diagnosis laboratory.

Taken together, all these arguments allow us to reclassify Leu351Arg as a likely pathogenic variant according to the ACMG classification. The use of multi-gene panels or whole-exome/genome sequencing as first line of genetic diseases diagnosis will bring out an increasing number of exonic and intronic *APOB* variants of uncertain significance in patients with either hypobetalipoproteinemia or hypercholesterolemia [37,38]. The cellular tool we propose in this work allows the global exploration of such variants. No *a priori* prediction of deleterious effects will be needed, since this all-in-one tool can be used to simultaneously explore the effects of epigenetics, splicing and post-transcriptional modifications [39] on apoB-100 secretion. Despite its high time consumption and due to the importance of functional studies for the primary HBL diagnosis, this tool will become a key determinant in routine diagnosis, allowing personalized hepatic follow-up in cases of *APOB* pathogenic variants.

## 4. Materials and Methods

### 4.1. Subjects, Biochemical and Genetic Analysis

The patients were recruited in Lyon (GENELIP/ASAP study; clinical trial registration number: NCT03939039, https://clinicaltrials.gov/ct2/show/NCT03939039, accessed on 3 October 2020). Written informed consent from the patients was obtained, according to French bioethical laws. The study was carried out according to The Code of Ethics of the World Medical Association (Declaration of Helsinki) and obtained the agreement of the ethical committee of the “Commission Nationale de l’Informatique et des Libertés” (CNIL) (N° 920,434).

Biochemical analyses were performed on Architect C16000 autoanalyzer (ABBOTT Diagnostics, Gurnee, IL, USA) for lipid profile (total TG, total cholesterol, HDL cholesterol and apoB). LDL-C was calculated using the Friedewald formula. Vitamin A, E and K concentrations were analyzed by HPLC (high-performance liquid chromatography) and vitamin D concentration was determined using a chemiluminescence immunoassay on IDS-ISYS (IDS-ISYS Tyne & Wear, UK).

Liver function was assessed by measuring plasma liver enzymes, including alanine aminotransferase (ALT), aspartate aminotransferase (AST) and gamma-glutamyl transpeptidase (GGT) on Architect C16000. ALT, AST and GGT were expressed as a multiple of the upper limit of normal (ULN). Patients with ALT > 1 ULN (>97.5th percentile) were considered to have liver injury. In addition, transient elastography was performed using FibroScan^®^ in order to determine: (1) the steatosis score, based on controlled attenuation parameter (CAP); and (2) the fibrosis score, based on liver stiffness measurement (LSM) [40]).

The DNA of HBL patients (LDL-C and apoB < 5th percentile, without any cause of secondary hypocholesterolemia) were sequenced as previously described [14].

### 4.2. Variants Selection

*APOB* sequence data were compared to the reference *APOB* sequence (GenBank accession no. NM_000384.2). We selected rare (allele frequency below 0.1% in GnomAD (https://gnomad.broadinstitute.org, accessed on 8 October 2020) VUS in *APOB* found in at least three relatives and showing vertical transmission. Variants’ predicted splicing effects were assessed bioinformatically using SPiP [41] and SpliceAI [42] and using Alamut Visual version v.2.14 (Interactive Biosoftware, Rouen, France), which incorporates predictions from MaxEntScan (MES), NNSplice, Splice Site Finder (SSF), GeneSplicer and Human Splicing Finder (HSF) and CADD [43].

### 4.3. Whole-Genome Sequencing

Whole-genome sequencing was performed by the Centre National de Recherche en Génomique Humaine (CNRGH, Institut de Biologie François Jacob, Evry, France). After a complete quality control, genomic DNA (1 µg) was used to prepare a library for whole-genome sequencing, using the Illumina TruSeq DNA PCR-free library preparation kit (Illumina Inc., San Diego, CA, USA), according to the manufacturer’s instructions. After normalization and quality control, qualified libraries were sequenced on a NovaSeq6000 platform from Illumina, as paired-end 150 bp reads. Libraries were pooled in order to reach an average sequencing depth of 30× for each sample. Sequence quality parameters were assessed throughout the sequencing run and standard bioinformatics analysis of sequencing data was based on the Illumina pipeline to generate FASTQ file for each sample. Raw sequence reads were aligned to the human reference genome (GRCh37) using BWA-MEM (version 0.7.5a) [44]. GATK 4.1.1.0 was used for indel realignment and base recalibration, following GATK DNA best practices [45] using the HaplotypeCaller tool and written to a VCF file, along with information such as genotype quality, strand bias and read depth for the SNV/Indel. Heterozygous variants predicted to affect protein coding or splicing, using SnpEff annotations (version4.3) [46], were kept during the analysis workflow. They were considered as rare when the minor allele frequency was lower than 1% in GnomAD. The copy-number variation analysis was performed using manta (v1.4.0) [47]. Filtering was performed using bcftools [48] and jvarkit [49].

### 4.4. Polygenic Risk Score

To study the polygenic cause of primary HBL, we used a compilation of 12 SNPs for patients from the family and controls (*n* = 856 subjects) developed by Talmud et al. [50]. Briefly, the genotypes of 12 SNPs were extracted from whole-genome sequencing data and the polygenic risk score (PRS) was calculated as previously described [3,50]. Patients with a PRS under the 10th percentile of controls were considered to have polygenic HBL.

### 4.5. Protein Modeling

ApoB modeling was performed with Phyre2 [51], based on the first 1000 residues of Uniprot apoB sequence (P04114), including silver lamprey lipovitellin. Impact of the missense variant was assessed with Missense3D [52]. Molecular graphs were generated with PyMol [53].

### 4.6. CRISPR-Cas9 Engineered Allelic Series

Huh7 cells (human hepatoma cell line, JCRB-0403) were maintained in Dulbecco’s modification of Eagle’s medium (DMEM) with l-glutamine (Gibco) containing 10% fetal bovine serum (FBS) and 1% antibiotic (10,000 U/L penicillin and 10,000 µg/mL streptomycin) in 5% CO2 at 37 °C and used as the parental cell line for genetic modification.

Mutants were created using an adapted CRISPR-Cas9 method [54] developed by Integrated DNA Technologies (Coralville, IA, USA (IDT)). In brief, guide RNAs were identified using https://benchling.com (accessed on 8 October 2020), crispor.tefor.net and IDT design tools https://eu.idtdna.com/pages/tools/alt-r-crispr-hdr-design-tool (accessed on 8 October 2020), selected by on-target score, off-target score and proximity to variant of interest. The Alt-R^TM^ two-part guide RNA (sgRNA) was formed using a crRNA XT recognition domain annealed with ATTO550 tagged tracrRNA transactivator domain (IDT, #1075927) and then complexed with the *S. pyogenes* CRISPR-Cas9 nuclease (IDT, # 1081058), following IDT recommendations. Single-strand oligos donors (ssODN) were designed with IDT DNA software, including the variant of interest and two synonymous variants, c.1059T > C and c.1066A > C, which are not predicted to affect *APOB* mRNA splicing, to avoid degradation by sgRNA itself. The ssODN were ordered with Alexa488 tag and phosphorothioate modifications at both 5′ and 3′ ends (see Appendix A).

The sgRNA-Cas9 nuclease complex and ssODN were transfected into Huh7 cells using Lipofectamine™ CRISPRMAX™ Cas9 reagent (ThermoFisher Waltham, MA, USA), following manufacturers’ instructions.

After 24 h, Atto550 + Alexa488 double-positive cells were sorted using flow cytometry (FACS ARIA II, Becton Dickinson, Franklin Lakes, NJ, USA) (see Appendix A) and transferred into 96-well plates for single-cell selection using limited dilution (seeding theoretically 0.75 cell/well). Typically, after 15–20 days, colonies were confluent enough to be harvested. Half of the cells were transferred to 24-well plates for further expansion and the remaining cells were used for DNA analysis. After DNA extraction with NucleoSpin Tissue kit (Macherey-Nagel, Düren, Germany), Sanger sequencing of *APOB* exon 9 was performed (Big Dye Terminator, ABI Prism, Thermo Fisher, Waltham, MA, USA) using the primers described in Appendix A to select clones carrying the variant of interest (see Appendix A). The absence of off-target editing and copy-number variants was confirmed by a next generation sequencing based-panel (NGS) [55].

### 4.7. ApoB-100 Quantification

Huh7 cells were grown in 25cm^2^ flasks with DMEM, 10% FBS and 2% DMSO (to be partially redifferentiated) for 3 days [56]. Medium was removed and cells washed with PBS. To promote lipid loading of apoB100 containing lipoprotein, cells were incubated with 0.6mM of oleic acid, complexed to BSA, for 0, 2 and 6 h. Medium was harvested and cells were lysed (HEPES pH7.5 50 mM, KCl 1.0 M, MgCl_2_ 2 mM, Triton X100 0.5%), collected and sonicated (10s, three times). The amount of apoB-100 produced and secreted, readout of VLDL secretion, was measured by ELISA (kit Mabtech AB, ref. 3715-1H-6, Nacka Strand, Sweden) following manufacturers’ instructions. This assay is specific for apoB-100 and does not recognize apoB-48. The apoB-100 concentrations were normalized to the protein concentrations in the cell lysate determined with Pierce BCA Protein Assay Kit (ThermoFischer, Waltham, MA, USA).

### 4.8. APOB Expression

*APOB* gene expression was quantified by digital-droplet PCR (ddPCR). Huh7 cells were cultivated as described above for 6h. Total RNA was extracted with RNAqueous-4PCR kit (Ambion, Thermo Fisher Waltham, MA, USA). Reverse transcription was performed with Transcriptor High Fidelity cDNA Synthesis Sample (Roche, Bâle, Switzerland) and a specific *APOB* amplicon (exon 17–18) was amplified and quantified by ddPCR on a QX200 ddPCR (BioRad, Hercules, CA, USA) using EvaGreen Supermix (BioRad). Primers are described in Appendix A and *B2M* used as internal control.

### 4.9. Statistics and Analysis

Triplicates from individually treated cells were assessed for *APOB* expression and apoB-100 quantification. Point and bar charts were generated using Excel (Microsoft, Albuquerque, NM, USA). The height of the bars represents the mean value of the data +/− standard deviation (SD). P values were calculated by using R’s two-tailed Welch’s *t*-test.

## 5. Conclusions

In summary, we provided consistent in vitro evidence that a novel missense variant in the βα1 domain of apoB is responsible for FHBL. Our data suggest that the Leu351Arg variant found in *APOB* results in a decrease in apoB-100 production both in cells and in medium, although future studies are needed to elucidate the molecular mechanisms through which apoB amino acid change leads to premature degradation. Finally, our study shows the utility of CRISPR/Cas9 engineering in characterizing functionality and reclassifying novel *APOB* VUS according the ACMG guidelines.

## Figures and Tables

**Figure 1 ijms-23-04281-f001:**
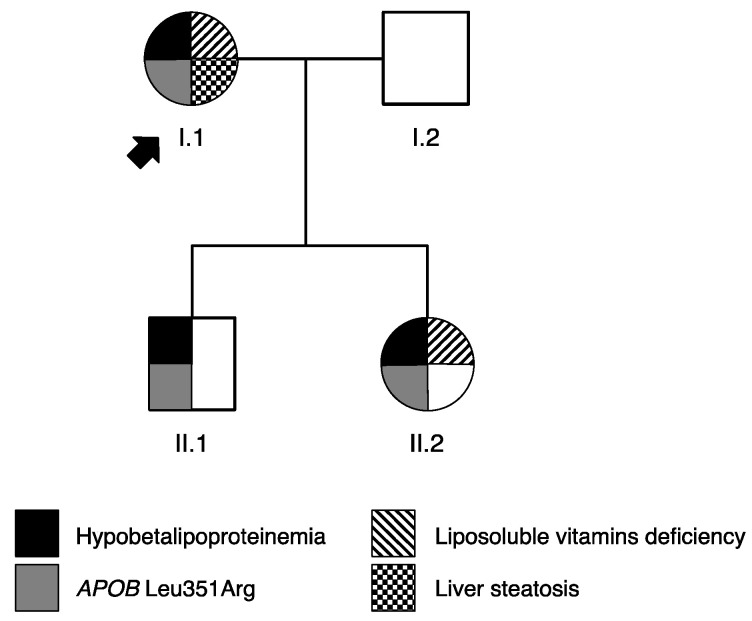
Familial pedigree tree. The squares indicate male family members and the circles female family members. The proband is indicated by a black arrow. The numerals below each symbol indicate individual family members.

**Figure 2 ijms-23-04281-f002:**
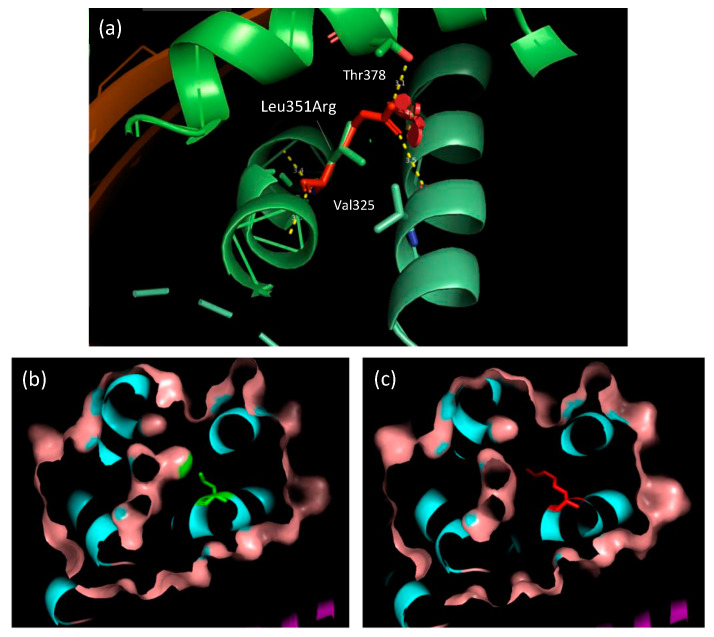
Location of the Leu351Arg missense variant in the three-dimensional structure of apoB βα1 domain. The substitution is located in an α-helix. The wild-type leucine residue is in green and the arginine variant is in red. (**a**) Two H-bonds are predicted between Arg351 and both Thr378 and Val325. (**b**) Visualization of the wild-type cavity volume. (**c**) Contraction of the cavity volume induced by the substitution.

**Figure 3 ijms-23-04281-f003:**
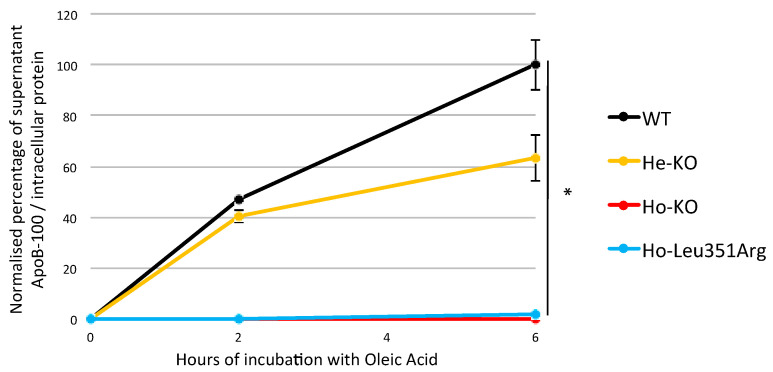
ApoB-100 secretion by Huh7 cell lines. Wild-type (WT) cells and stable cell lines carrying FHBL causative variant in heterozygous state (He-KO) or homozygous state (Ho-KO), or carrying variant of interest in homozygous state (Ho-Leu351Arg) were cultured with oleic acid (OA) 0.6 mmol/L, complexed to BSA, for 0, 2 and 6 h. The apoB-100 concentrations were measured in the medium by ELISA and were normalized to the protein concentrations in the cell lysate. The WT cell line 6 h after OA incubation was considered as the reference for data normalization. * *p* < 0.05 (Welch’s *t*-test). Error bars, ±S.D. (*n* = 3).

**Figure 4 ijms-23-04281-f004:**
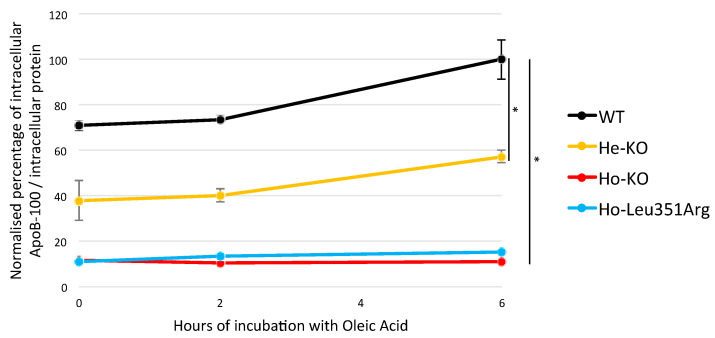
Estimation of apoB production by Huh7 cell lines. Wild-type (WT) cells and stable cell lines carrying FHBL causative variant in heterozygous state (He-KO) or homozygous state (Ho-KO), or carrying variant of interest in homozygous state (Ho-Leu351Arg), were cultured with oleic acid (OA) 0.6 mmol/L, complexed to BSA, for 0, 2 and 6 h. The apoB-100 concentrations in the lysate were measured by ELISA and were normalized to the protein concentrations in the cell lysate. The WT cell line 6 h after incubation with OA was considered as the reference for data normalization. * *p* < 0.05 (Welch’s *t*-test). Error bars, ±S.D. (*n* = 3).

**Figure 5 ijms-23-04281-f005:**
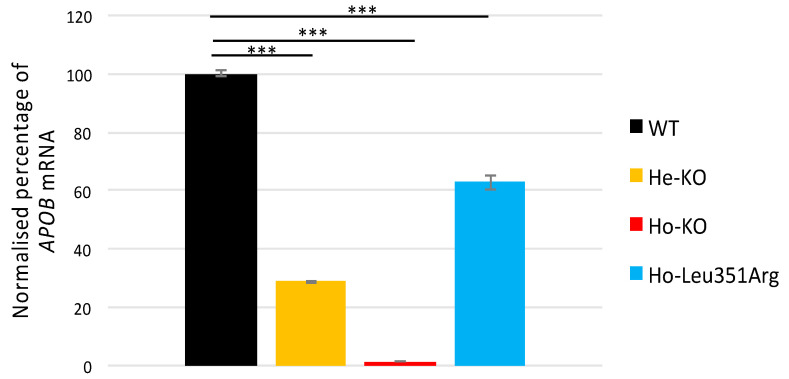
Relative quantification of *APOB*-100 mRNA produced over 6 h by Huh7 cell lines. Huh7 cells were cultivated as described above for 6 h. Total RNA was extracted, reverse transcription was performed and a specific *APOB* amplicon (exon 17–18) was amplified and quantified by digital droplet PCR (ddPCR). The wild-type (WT) cell line was considered as the reference for data normalization for cell lines carrying FHBL causative variant in heterozygous state (He-KO) and homozygous state (Ho-KO), or carrying variant of interest at homozygous state (Ho-Leu351Arg). *** *p* < 0.001 (Welch’s *t*-test). Error bars, ±S.D. (*n* = 3).

**Table 1 ijms-23-04281-t001:** Paraclinical results in proband and her family.

Individuals	I.1	II.1	II.2
Sex	F	M	F
Age range (year)	50–60	30–40	20–30
TG (mmol/L)	2.19 (↑)	1.36	0.44
Total cholesterol (mmol/L)	3.40 (↓)	3.43 (↓)	2.61 (↓)
HDL-c (mmol/L)	1.11	1.54	1.47
LDL-c (mmol/l)	1.30 (↓)	1.27 (↓)	0.94 (↓)
ApoB-100 (g/L)	0.45 (↓)	0.39 (↓)	0.23 (↓)
TC/apoB	2.92	3.4 (↑)	4.38 (↑)
AST (ULN)	0.95	1.11	0.60
ALT (ULN)	0.88	1.78	0.45
GGT (ULN)	2.22	0.30	0.42
Vit A (µmol/L)	2.92	2.93	1.77
Vit D (µmol/L)	19 (↓)	85	44
Vit E (µmol/L)	15.2 (↓)	21.7	15.6 (↓)
Vit K1 (ng/L)	94	NA	NA
Prothrombine time	100%	100%	97%
Liver elastometry:			
CAP (dB/m) (steatosis score)	359 [S3]	NA	NA
LSM (kPa) (fibrosis score)	5.5 [F0–F1]	5.3 [F0–F1]	5.5 [F0–F1]

TG: triglyceride, HDL-c: high-density lipoprotein cholesterol, LDL-c: low-density lipoprotein cholesterol, TC: total cholesterol, AST: aspartate aminotransferase, ULN: upper limit of normal, ALT: alanine amino-transferase, GGT: gamma glutamyl transpeptidase, CAP: controlled attenuation parameter, LSM: liver stiffness measurement, ↑: increased, ↓: decreased, NA: not available.

## Data Availability

Data are available upon reasonable request. All data relevant to the study are included in the article or uploaded as Appendix A. Patient data relevant to the study are included in the article. Further experimental data are available from Xavier Vanhoye (xavier.vanhoye@chu-lyon.fr) upon reasonable request.

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
