# Peer review of "APOB CRISPR-Cas9 Engineering in Hypobetalipoproteinemia: A Promising Tool for Functional Studies of Novel Variants"

_ijms, 2022, doi:10.3390/ijms23084281_

Round 1
Reviewer 1 Report
The authors focus on one newly discovered missense mutation (Leus351Arg) in the apolipoprotein B (APOB) gene in familial hypobetalipoproteinemia (FHBL). The authors examine whether the mutation is the direct reason in HBL. Using CRISPR/Cas9 system, the homozygous mutation (Leus351Arg) in the APOB gene generates in Huh7 cells. The stable cell lines with homozygous mutation lead to a decrease in the APOB-100 secretion and production and the APOB-100 expression. These results show the usefulness of the technique to analyze the role of the novel mutation in APOB gene in HBL.
The authors identify the novel mutation in the APOB gene in FHBL by sanger sequencing and clearly show the effect of this mutation in the APOB-100 production in the cell lines that have the homozygous mutation by the CRESPR/ Cas9 system.
- In Figs. 3 and 4, the authors show the secretion and production of APOB-100 in cells with homozygous mutation. The author should show the effect of biological effects on these cells, for example, biochemical quantification of triglyceride and/or cholesterol.
- In Fig. 5, the authors show a decrease in the expression level of APOB-100 mRNA in cells with homozygous mutation. The author should discuss the reason of the decrease the APOB-100 expression of 40% in cells with homozygous mutation as compared with the wild type.
- The authors missed the spelling: line 181: Hz-Ko.
Reviewer 2 Report
The Authors described the original method describing APOB CRISPR-Cas9 engineering in hypobetalipoproteinema. The paper described the procedure well. I would suggest to also include figures of the results mentioned in the paper such as FACS plots of the results section.
Round 2
Reviewer 1 Report
I find the revised paper acceptable.